# Applicability and potential of monitored reconstruction in computed tomography

**Marat Gilmanov**[1,2]*, **Konstantin Bulatov**[1,3], **Oleg Bugai**[1,2], **Anastasia Ingacheva**[1,2], **Marina Chukalina**[1,3], **Dmitrii Nikolaev**[1,3], **Vladimir Arlazarov**[1,3]

**1** Smart Engines Service LLC, Moscow, Russia, **2** Institute for Information Transmission Problems RAS, Moscow, Russia, **3** Federal Research Center Computer Science and Control RAS, Moscow, Russia

* m.gilmanov@iitp.ru

**Data Availability Statement:** All relevant data are within the paper.

**Funding:** No external funding was received for this study.

## Abstract

Monitored tomographic reconstruction (MTR) is a potentially powerful tool for dose and time reduction in computed tomography scanning. We are the first to study the issue of practical implementation of MTR protocols in current-generation real-life instruments. We propose an empirical quantitative model for calculating acquisition and reconstruction times. It is demonstrated that projection acquisition order has a significant impact on the time and dose of tomographic experiments. The new alternative acquisition most suitable for MTR protocols is proposed. To estimate the restrictions and scope of applicability for MTR four typical commercial setups are studied within a proposed model. We construct an experimental stand for achieving a real-time reconstruction, together with validation of the proposed acquisition time model. We demonstrate that real-time reconstruction may be implemented without slowing down an acquisition process. An optimization of reconstruction from partial data is proposed, which allowed the production of 385 and 440 reconstructions for standard and proposed acquisition orders correspondingly during a single acquisition of 512 projections. The results of the study demonstrate that with proposed optimizations MTR can be effectively utilized for practical applications using the current generation of existing setups in industrial and nano tomography fields.

## Introduction

Computed tomography (CT) is a non-invasive method for exploring the inner structure of various objects. CT is actively used in various fields of applications, with a variety of setups and specific features in its design. Four of the most common classes of CT setups with fundamental differences between each other can be distinguished: medical CT [1], cargo CT [2], nano CT [3, 4] and industrial CT [5, 6]. In the scope of the present work, the discussion would naturally arise around these four classes of setups, with a very wide class of industrial CT being represented as micro CT [5, 7].

An eternally pressing problem for CT has always been the X-ray dose reduction for an object under study [8]. Various approaches and methods are utilized and effectively combined to achieve significant results [9, 10]. Even if the method could provide only a minor

**Competing interests:** NO authors have competing interests.

improvement, it is still can be included in CT pipelines to play its part, which is why no dose reduction methods can be ignored. The very close problem to dose reduction is a total study time reduction. This particular issue is especially important for industrial and nano CT experiments, which may take up to several days [4, 6, 11].

One of the new promising concepts, allowing to achieve significant dose and time reductions, is a Monitored Tomographic Reconstruction (MTR) [12, 13]. The main idea behind MTR is the production of multiple reconstructions on a real-time scale as soon as new data is acquired. These reconstructions from partial data allows to estimate the quality of CT images, and further decide if the data acquisition can be stopped. The stopping could occur either based on general quality assessment [12] or based on accuracy estimations of reaching specific goals the CT experiment is conducted for [14].

While the concept of MTR seems to be quite attractive, even a single CT reconstruction under some conditions may take longer than the *entire data acquisition process*, which raises the concern if the MTR even has scope of applicability. At the same time, all studies in this field up to now are directed towards the issue of dose reduction [3, 12, 13, 15, 16], based on numerical experiments under the presumption of instant reconstruction and instant quality assessment, even in the cases when real-time reconstruction is unattainable by design [14].

The main question discussed in the present paper is whether the MTR concept could be implemented in current-generation setups or if it would require the development of next-generation hardware. The applicability of MTR is studied for typical CT setup configurations in four different fields of application. The contributions of this work may be formulated as follows:

1. A universal quantitative model is proposed for MTR dose and experiment time estimation under different projection acquisition protocols.

2. The requirements and restrictions for implementation of MTR protocols for currently existing CT setups are formulated based on the proposed model.

3. A new protocol based on specific projection acquisition order is proposed, which demonstrates promising properties regarding the MTR protocol.

4. The requirements for the reconstruction algorithms are formulated and simple optimizations are proposed for known algorithms under the conditions of MTR protocols.

## Monitored tomographic reconstruction

To clarify the following experiments and discussion let us start by re-introducing the concept of MTR in more detail. MTR is a type of projection acquisition protocol, based on the reconstruction in real-time scale as the new projection data becomes available [12]. As the new subset of data is acquired, the reconstruction from currently available partial data is performed (*partial reconstruction*). The produced partial reconstructions are further utilized for quality assessment based on the task the examination is conducted for [12, 14]. Each assessment produces a *decision point* in the history of quality estimation. This history and characteristics of the changes in the task solution accuracy allow making the decision to continue or to stop the projection acquisition process, based on some prior knowledge about the task and fine-tuning of calibrations according to *stopping rules*. Early interruption of data acquisition allows to reduce the total number of projections, and hence the total dose and time of an experiment. The principal scheme of acquisition based on MTR protocol is summarized as a flowchart in Fig 1.

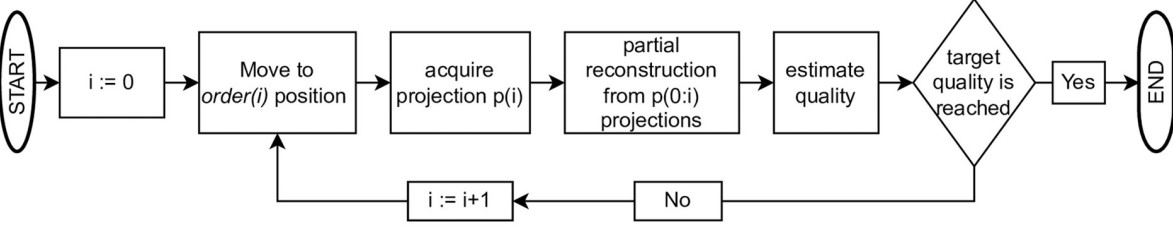

**Fig 1. Flowchart of MTR protocol.**

From the discussion above one could notice that the most opaque problem is the quality assessment, since the quality will always be lower than if the experiment would continue to the end, which results in a trade-off optimization problem. If the task for the CT experiment is formulated as "obtain the best quality no matter the cost" (which is often the case in micro and nano CT), the MTR will likely be useless. However, most practical investigations are conducted to achieve a certain goal with the minimum dose possible, e.g. quantitatively estimate the lung lesions in medical CT or to investigate and quantify defects in industrial CT. In such cases, MTR allows to start of an automatic process of task solving (such as detection or segmentation) during the process of the acquisition [14]. The decision will be made based on the dynamics of the obtained solution characteristics, i.e. when the answer starts to change only "a little", it is unlikely for the solution to change anymore and the acquisition may be safely stopped. In the original MTR papers [12, 13] it was shown, that even for a general task of obtaining reconstruction with "good enough" quality with a lower dose, the stopping rule can be constructed to achieve a mean dose reduction compared to fixed-dose protocols. These modeling results demonstrate a significant potential of MTR protocols for dose reduction. However, it will be demonstrated below that implementing MTR creates certain restrictions and requirements for both hardware and software components of CT setups.

The quality assessment problem is vast, and with the rise of neural network models the image quality understanding could be changed even more if the reconstruction image is used only by the corresponding artificial intelligence system. For example, some specific tasks in Robo CT [17] require less than 20 projections to be completed, while in nano CT 1000 projections could be not enough [4]. While these are extreme cases, they show the range of understanding the image quality in different areas which is why we further discuss the image quality in a somewhat general sense.

The necessity for an acceptable reconstruction quality creates certain restrictions on the acquisition order. The meaning of acquisition order is best illustrated by the Eq 1:

$$order(i) = f(i),$$

$$i, order(i) \in (0, N_p - 1),$$

$$angle\_values(j) = a_{min} + j/N_p \times (a_{max} - a_{min}),$$

$$angle(i) = angle\_values(order(i)).$$

(1)

Here $a_{max}$, $a_{min}$ is an angle range, $i$ is the sequential index of captured projection, and $N_p$ is a total number of projections. In the presented formula the $order(i)$ is a permutation over the indexes in range $(0, N_p - 1)$. In current-generation CT setups, the order of projections capturing is *always* consecutive (incremental), which corresponds to the case of $order(i) = i$ (Fig 2 (a)). Although for consecutive order the acceptable reconstruction results occur only at the

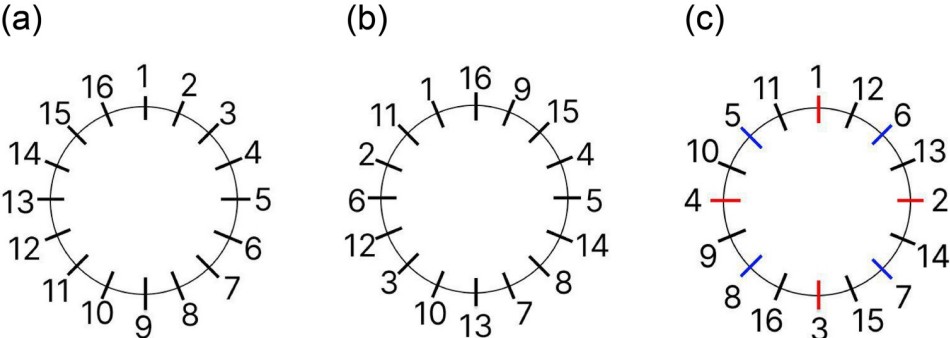

**Fig 2. Schematic illustration of acquisition order for $N_p$ = 16.** The position of strokes corresponds to a projection angle *angle*($i$) and numbers to a sequence number $i$ (Eq 1). (a) Classical case of consecutive order, (b) random or arbitrary order, and (c) logarithmic order with $N_{prescan}$ = 4, the same colors correspond to projections that are obtained in the same rotation.

very last stages of acquisition if the classical fast reconstruction algorithms such as FBP (Filtered Back Projection [18]) or FDK (Feldkamp, Davis and Kress algorithm [19]) are used (Fig 3 upper row). In the original theoretical papers [12, 13], as a workaround for this problem, it was proposed to utilize a "random" order (Fig 2(b)). More generally random order could also be thought of as a "smart" order when the next projection angle is chosen based on the current partial reconstruction [20] or field of view conditions [21], which could also be profitable for the MTR protocols.

On the other hand, while random order looks profitable from a theoretical point of view (Fig 3 middle row), it may become a burden in experimental implementations of MTR. This is due to the fact that the time required for rotation per projection for random order is significantly higher than for consecutive order. It can be estimated that an average angular distance for random order is $d(i) \approx 90$ deg, if the rotation is possible in both directions and $d(i) \approx 180$ deg if the rotation is possible in only a single direction, i.e. gantry rotation in medical CT. This means that a total angular distance is equal $d_{total} = 90 \times N_p$ and $d_{total} = 180 \times N_p$ respectively, while in case of consecutive order $d_{total} = 360$ independent from $N_p$. Since the rotation speed is always limited and for many applications is comparable with the projection acquisition speed, the experiment may become much slower if the random order is chosen. Moreover, most setups would not allow disabling or screening the radiation during rotation, which would also lead to an increase not only in total time but also in full dose.

In the present paper, we suggest to use "logarithmic" order as an alternative to consecutive and random orders. This order could be described as follows: first, a full rotation with $N = N_{prescan}$ projections is performed; after that acquisition is performed with $N = N_{prescan} \times 2^i$, $i = 0, 1, \ldots$ per rotation (Fig 2(c)). The main advantage of the proposed angles order is that after each full rotation, the acquired projections are equidistant by angle, which allows utilization of fast and simple classical reconstruction approaches without a quality loss typical for random order (Fig 3). At the same time, total angular distance can be calculated as follows:

$$d_{total} = 360 \times (1 + log_2(N_p/N_{prescan})) \tag{2}$$

This way the dependency of $d_{total}$ from $N_p$ is logarithmic, and from the point of view of rotation time costs a compromise between previously introduced orders can be reached.

As it was shown above, the partial reconstructions in consecutive order obtained with the use of classical fast reconstruction algorithms would not be helpful for quality assessment (Fig

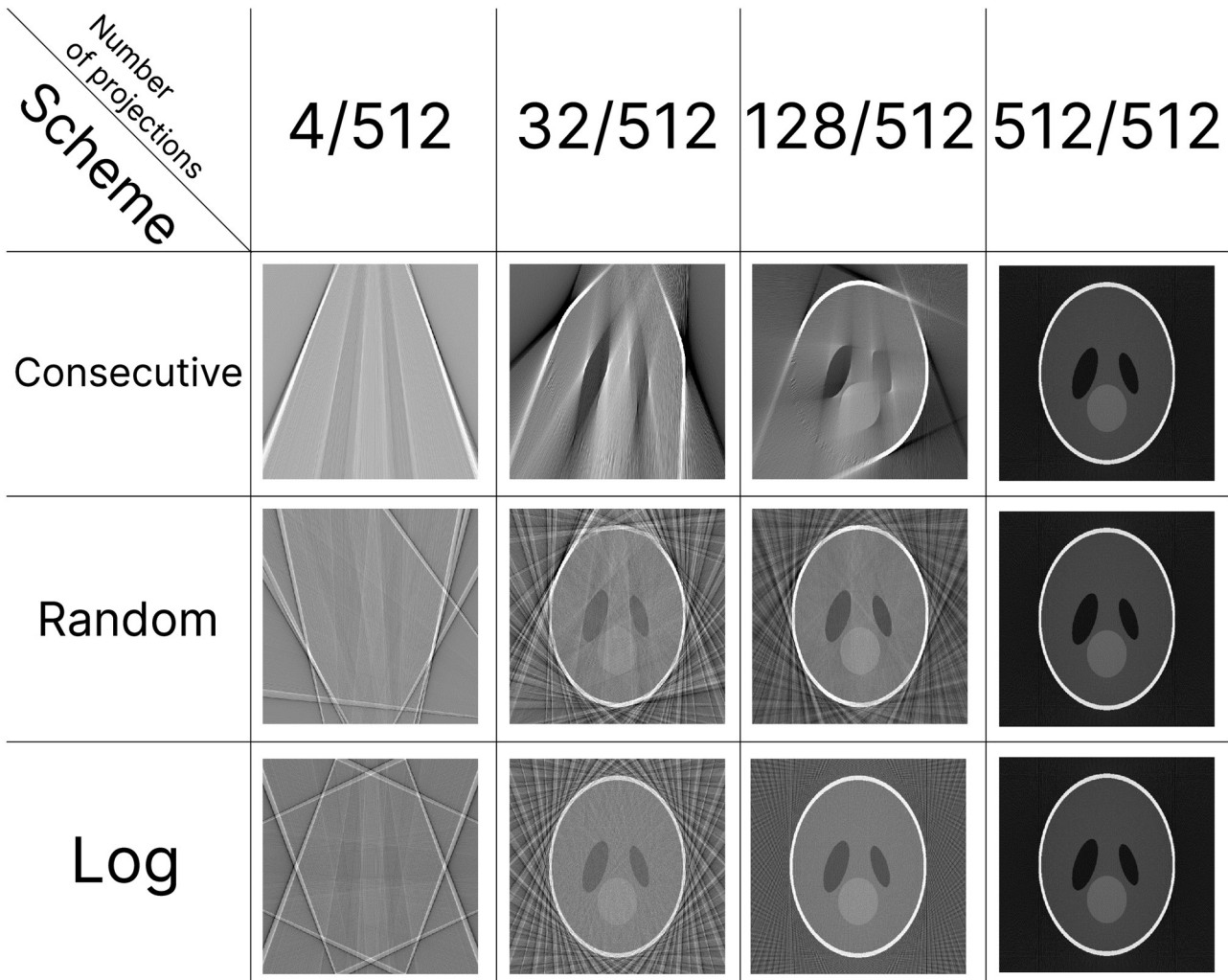

**Fig 3. Partial reconstructions in fan geometry obtained by FBP reconstruction algorithm for different numbers of available projections and for different acquisition orders.** The maximum total number of projections $N_p$ = 512 and image size 512 × 512 is the same for all cases. Gray scale limit values are the same for every column.

3), since the full range of equidistant projections is required in this case. Non-equidistant projections (random order) or limited angle range (consecutive order) lead to reconstruction corruptions demonstrated in Fig 3. These cases are usually dealt with by utilizing special slow iterative [22] or neural network-based methods [23]. At the same time, the reconstruction speed is a critical factor for MTR implementations and if not even a single reconstruction can be performed for the time of acquisition, no real-time reconstruction would be possible. On the other hand the number of partial reconstructions ($N_{rec}$) in total experiment time would define the "resolution" of MTR protocol on the stopping decision scale. In the extreme case of "instant" reconstruction, typical for theoretical studies, the maximum resolution would be reached with $N_{rec} = N_p$.

Thus the problem of utilizing slow and stable reconstruction algorithms [24] such as SIRT (Simultaneous Iterative Reconstruction Technique [25]), requires close attention and it is directly connected with the potential of utilizing consecutive and random scanning orders in

the MTR framework. It should also be noted, that in a particular implementation of MTR protocols, the experiment time could be artificially increased for higher resolution if necessary, although such measures would almost always be unwanted. In fact, any deceleration would lead to a direct increase in total experiment time in the worst-case scenario when the stopping rule conditions would never be met. For this reason, we aim to study if it is possible to implement MTR protocols without a significant increase in total dose and time in relation to standard protocols for particular CT setups. We will consider the implementation of the MTR protocol to be a software extension under existing setup tools. This way, the reconstruction algorithm should be able to produce a partial reconstruction in time comparable with the acquisition of single (or at least a batch of several) projections. At the same time, the reconstruction algorithm should allow utilization of the arbitrary angle range, which in some sense produces controversial requirements.

## Analysis of MTR implementation for commercially available setups

### Data acquisition

Within this section, we will consider the possibility of MTR implementation on existing commercially available setups, by modifying only the software controlling part. The principal mechanism for dose reduction in MTR is the reduction of the number of projections by interruption of acquisition, which may potentially happen in arbitrary decision points [12]. For MTR to achieve a positive effect a very strict requirement may be formulated: the implication of MTR in the worst-case scenario should not increase the dose in respect to a standard protocol. While this does not constitute a necessary condition it is a sufficient condition, unrelated to the amount of average projection count reduction, which may vary based on a lot of factors such setup configuration or the quality assessment method itself. Although the quantitative estimations of the amount of dose reduction are out of the scope of the present paper, further proposed numerical models may be used to assess the effectiveness of MTR protocols based on average projection number reduction.

From the argumentation above one could assume, that MTR could be replaced by simply reducing the average projection number in standard protocols, although there are principal differences between the two approaches that are easy to demonstrate by an example. Let us consider the task of evaluating the amount of affected tissues during CT examination of the lungs [14]. In this case, if the affection is present, the study has to be the best quality for precise estimation of the affection area. At the same time, if there is no affection at all, the acquisition should be canceled as soon as possible. However, it is not known in advance if the affection would be observed and all the patients are scanned assuming the worst-case conditions. In this case, an effectively implemented MTR protocol should be able to greatly reduce the dose for healthy patients, and the average dose reduction would depend on the relative ratio of healthy and ill patients.

To evaluate if the previously formulated requirements can be met, we will further consider the factors contributing to the total dose and time of the CT experiment. On the base of this consideration, the empirical model will be constructed and CT setups typical for the main CT application areas will be assessed.

It should also be noted that under some simplification dose is proportional to a total object X-ray exposition (which is not true mainly in the case of the non-trivial time dependency of radiation intensity). It allows us to simplify the comparison by using the same units for time and dose, i.e. frame and object expositions. Moreover, in a number of practically important cases, the dose (measured in time units) would coincide with total study time.

Based on experience with the micro CT setup we have direct access to, we were able to derive the main factors contributing to time and dose in an experiment and to generalize these factors into an empirical model suitable for describing arbitrary CT setup. Two main factors are the expenses for projection acquisition $T_{exp}$ and expenses associated with rotation system $T_{move}$. The acquisition time could be further divided into useful exposition $T_{frame}$ and constant overheads $C_{det}$ of projection acquisition independent from the exposition. Expenses associated with the rotation system in a general case depend on total projection count $N_p$, acquisition order $order(i)$, and rotation angular speed. Similar to exposition there also can be constant overheads for movement stabilization $C_{move}$, i.e. start/stop of rotation. In the simplest case, the total time/dose is calculated as the sum of $T_{exp} + T_{move}$. Although in some cases (medical or cargo CT) rotation and exposition are conducted simultaneously, which means total time is defined as $max(T_{exp}, T_{move})$, and optimally these times should be chosen the same. The last factor to assess is the possibility of the use of pulse X-ray sources or X-ray blocking gates. Previously considered factors contributed equally to time and dose under the condition of continuous X-ray source. Using of pulse source or blocking gates allows to reduce dose by radiating objects only for directly helpful exposition, which, however, does not reduce or even increase total experiment time. It should be noted that expenses for gating in general case could contribute differently in dose and time ($C_{gate\_dose}$, $C_{gate\_move}$), since gating may be synchronized with rotation, in which case contribution to the total time of $T_{gate}$ will be zero.

Based on conducted consideration the following model could be proposed:

$$T_{exp}^i = T_{frame} + C_{det}$$

$$T_{move}^i = C_{move} + dist(i, i+1)/s_{rot}$$

$$T_{gate}^i = gate\{0, 1\} \times C_{gate\_move}$$

$$D_{exp}^i = T_{exp}^i$$

$$D_{move}^i = gate\{0, 1\} \times T_{move}^i$$

$$D_{gate}^i = gate\{0, 1\} \times C_{gate\_dose}.$$

(3)

With quasi-stationary conditions:

$$T_{total} = \sum_{i < N_p} T_{exp}^i + T_{move}^i + T_{gate}^i$$

$$D_{total} = \sum_{i < N_p} T_{exp}^i + D_{move}^i + D_{gate}^i,$$

(4)

and continuous movement conditions:

$$T_{total} = \sum_{i < N_p} max(T_{exp}^i + T_{gate}^i, T_{move}^i)$$

$$D_{total} = \sum_{i < N_p} T_{exp}^i + D_{gate}^i.$$

(5)

Here $dist(i, i-1)$ is an angular distance between neighboured projections, which depends on $order(i)$ (Eq 1), and $s_{rot}$ is rotation or angular speed.

Let us highlight some properties of the proposed model considering the dependence of dose and time related to the total projection number. The main factors that affect total

experiment time and dose are the exposition, angle order, and gating, while most of the others may be thought of as aggregated overheads which are similar to exposition constant per projection or linear by $N_p$. It is clear that different asymptotic behaviour may occur based on the order of projections acquisition. Although it is interesting to consider not only asymptotic behavior but also the relative contributions of rotation expenses compared to a helpful exposition. It should also be noted that in the case of gating present, the dependency on acquisition order would be completely negated.

For numerical studies 4 classes of CT setups were chosen: micro CT, nano CT, medical, and cargo CT. For each area parameters for modeling were chosen from ranges typical for standard setups in these areas. Parameters for micro CT were measured on commercially available setup produced by ELTEH-Med company [5]. Parameters for nano [3], medical [26], and cargo [2] CT were estimated based on the data available in the literature (Table 1). For parameters with a valid range of values, a single intermediate value was chosen for the simplification of results representing. Zero values in Table 1 correspond to values unavailable from literature and that are presumably negligible. It should be recalled, that while some parameters could be optimized by setup modifications, here we consider the existing setups with known configurations.

Apart from parameters used for modeling by Eq 3, the Table 1 also contains the modeling results under the standard protocol of acquisition, which demonstrates the principal difference of setups in timescale. Also, Table 1 contains the typical detector sizes, which will further be used to estimate the reconstruction times. For our estimations, helical scanning trajectory is hereby replaced by corresponding circular cone beam CT with displace for unification.

The proposed model 3 allows to obtain the dependencies of the total time and dose of the experiment from the number of acquired projections. Graphs on Fig 4 represent the result of such modeling for all chosen configurations of setups and for different acquisition orders: consecutive, random, and logarithmic (see previous section). As it was pointed out previously, the dose is represented in time units corresponding to an object exposure.

**Table 1. Parameters of particular CT setups, corresponding to different areas of application.**

| Parameter | micro CT | nano CT | medical CT | cargo CT | EBCT |
|---|---|---|---|---|---|
| Model of CT setup, and | ELTEH-Med | In-lab tool | SOMATOM | Large scale | Cardiac 3-D |
| source of information | [5] | [3] | 16-slice [26] | MV CT [2] | Densitometer [27] |
| Total angles ($N_p$) | 512 | 800 | 720 | 720 | 400 |
| Sample rotation angles range | 360 | 180 | 360 | 256 | 360 |
| Detector size ($N_x \times N_z$) | $2048 \times 2048$ | $2048 \times 2048$ | $736 \times 16$ | $1360 \times 1$ | $720 \times 2$ |
| Frame exposition, s. ($T_{frame}$) | 1 | 180 | 0.00138 | 0.0833 | $7 \times 10^{-5}$ |
| Time per rotation, s. ($1/s_{rot}$) | 60 | 60[2] | 1 | 60 | 0[3] |
| Detector overheads, s. ($C_{det}$) | 1 | 1[2] | 0[1] | 0[1] | 0[1] |
| Motion overheads, s. ($C_{move}$) | 0.1 | 0.1[2] | 0[1] | 0[1] | $1 \times 10^{-5}$[3] |
| Gating ($gate\{0, 1\}$) | + | − | − | − | + |
| Gating overheads, s. ($C_{gate\_dose}$) | 0.1 | − | − | − | 0[1] |
| Continuous movement | − | − | + | + | − |
| Total time, consecutive order, Eqs 7 and 8, s. ($T_{total}$) | $1.1 \times 10^3$ | $4.9 \times 10^4$ | 1 | 60 | 0.03 |

The parameters of particular CT setups used for modeling by Eq 3.

[1] data is unavailable and contribution is supposedly negligible,

[2] data is unavailable and supposed to be equal to the corresponding parameters from the micro CT case,

[3] for EBCT the time overheads for readjustment of projection angle should be independent of angular distance and hence is considered to be constant.

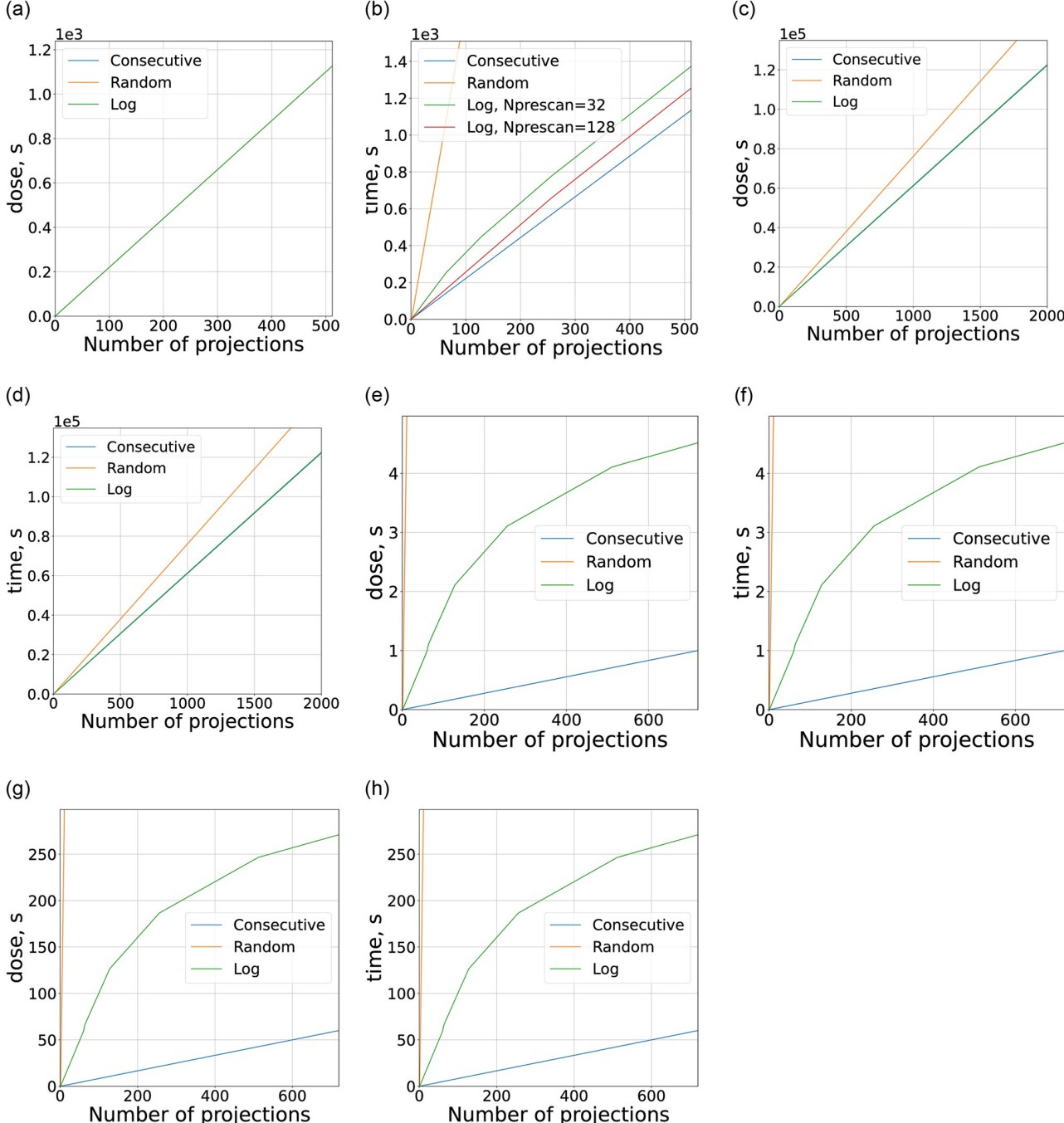

**Fig 4. Dose and experiment time dependence from the number of projections acquired with fixed maximum projection number $N_p$ based on model Eq 3 and with parameters from Table 1.** The dose is measured in object exposition time units for convenience of comparison. (a) Dose, micro CT. (b) Time, micro CT. (c) Dose, nano CT. (d) Time, nano CT. (e) Dose, medical CT. (f) Time, medical CT. (g) Dose, cargo CT. (h) Time, cargo CT.

Let us start with the first two pairs of graphs related to micro and nano tomography (Fig 4a–4d). While dose here is not as important as in medical CT, it is still critical in some particular applications such as a study of biological objects or jewelry. Under the influence of X-ray radiation, irreversible changes may occur in a biological object, while gems and jewelry may

change their shades of color which will decrease their cost. The dependencies for micro CT were modeled with gating, which as expected allows to remove the dose dependency from acquisition order. Although the effects of acquisition order on total experiment time is quite significant and utilizing random order leads to a slowdown of $\approx 10$ times (orange line on ([Fig 4e and 4f])), which would be unacceptable for most of the practical applications. At the same time for nano CT ([Fig 4c and 4d]) the main contribution is defined by the long exposure times and the effects of angle order are quite insignificant. Another point to be made is related to the behavior of logarithmic order dependencies. The total time can be decreased by increasing the $N_{prescan}$ parameter. This way the number of prescan projections becomes a convenient control parameter for achieving a trade-off between consecutive and random order.

For the cases of medical and cargo CT, the dependency from acquisition order is even more pronounced and the dose and time increase seems to be critical even for high $N_{prescan} = 128$ ([Fig 4e and 4h]) ($\approx 1s$. for medical CT). This means that the implementation of MTR in currently available medical and cargo CT setups could be limited to consecutive order. It seems to be necessary to develop conceptually new approaches that would negate the severe slowdown introduced by arbitrary acquisition orders before the MTR could be effectively implemented in this type of setup. Moreover, the industry standard in medical CT is the spiral CT acquisition, for which the stopping task is even less trivial since the same projections can first for some layers and last for others [14].

## Real time reconstruction

The next important aspect of MTR is real-time reconstruction, which is directly related to the question of the applicability of various reconstruction algorithms. It should be noted that acquisition, reconstruction, and quality assessment for stopping rules are the processes that do not block each other and could be performed asynchronously and simultaneously. In the context of MTR, it is only worth considering the asynchronous scheme, since otherwise the acquisition would become significantly slower. As it was noted previously, the time required for obtaining the partial reconstruction defines the resolution $N_{rec}$ of MTR protocol on the scale of stopping decisions. It should be clear that the faster the reconstruction process is, the more detailed quality assessment history one would be able to retrieve, and a more precise stopping point position would be determined. In the extreme case of "instant" reconstruction, the number of partial reconstructions, and hence stopping points decisions, would be equal to the total number of projections $N_{rec} = N_p$. In this sense, the upper limit to the resolution is reached at the condition of maximum partial reconstruction time being less than the time of acquisition of a single projection (in general case the partial reconstruction time may depend on the number of currently obtained projections). Although such an upper limit is not necessarily required the sufficiency criteria should be defined based on solving tasks, chosen methods of quality assessment, and stopping rules. From the general sense, it is also clear, that the lower limit of resolution for MTR to be possible at all is $N_{rec} > 3$: the first partial reconstruction provides a baseline, the second provides the difference with baseline and the third one allows to make a decision based on compared differences, although the last partial reconstruction should be finished before the acquisition is finished, which produce strict inequality.

To discuss the applicability of the MTR protocol it is essential to consider the typical reconstruction to acquisition times relation. We will further consider the same four cases as in the previous subsection. Reconstruction time estimations are based on the parameters of a typical GPU unit of one of the latest generations ([Table 2]). The model accounts for data loading

**Table 2. Parameters of GPU-based computing unit related to the proposed model.**

| Parameters of Nvidia GTX 4090 GPU | Values |
|---|---|
| Floating point operations per second, TFlops/s. ($s_{rec}$) | 85.2 |
| Data loading speed throughout PCIe interface, MB/s. ($s_{load}$) | 500 |
| Data access speed, TB/s. ($s_{memory}$) | 1 |
| The number of operations per voxel reconstruction. ($P_{voxel}$) | 600 |
| The number of operations per pixel preprocessing. ($P_{pixel}$) | 200 |
| GPU Random Access Memory (RAM) size, Gb | 24 |

The GPU parameters together with related parameters in a model (Eq 6)

speed, data access speed, operations per second, and some empirical meta-parameters estimated from actual software.

The following model is proposed:

$$proj\_size = N_x \times N_z \times N_a$$

$$vol\_size = N_x \times N_x \times N_z$$

$$T_{load} = proj\_size / s_{load}$$

$$T_{memory} = proj\_size \times / s_{memory} \qquad (6)$$

$$T_{prep} = proj\_size \times P_{pixel} / s_{rec}$$

$$T_{rec} = it\_count \times vol\_size \times N_a \times P_{voxel} / s_{rec}$$

$$T_{total} = T_{load} + T_{memory} + T_{prep} + T_{rec}$$

Here $N_x$ and $N_z$ are the numbers of columns and rows of a detector, $N_a$ is a number of projections used for reconstruction, $proj\_size$, and $vol\_size$ are the total numbers of pixels on projections and voxels in the reconstructed volume. $T_{load}$ and $s_{memory}$ is the time and speed associated with loading of data from central processor to Graphical Processing Unit (GPU) memory, $T_{memory}$ and $s_{memory}$ is the time and speed associated with data access on GPU, $T_{prep}$ and $T_{rec}$ are times of preprocessing and reconstruction with corresponding empirical parameters $P_{pixel}$ and $P_{voxel}$ of operations per data unit with $s_{rec}$ number of operations per second. In the proposed model the reconstruction time is estimated based on contributions from reading of data, loading data to GPU, projections preprocessing, and reconstruction itself, with the main contribution related to reconstruction. In the proposed model the loading of data to GPU memory is supposed to be performed only once, and GPU memory is considered to be large enough to store both volume and projection data. It should be emphasized that the main purpose of the proposed model is to produce an order of magnitude correct values while keeping it relatively simple and providing the possibility to asses different reconstruction algorithms. The empirical parameters $P_{pixel}$ and $P_{voxel}$ are estimated based on time performance measured on commercially available software SmartTomoEngine (STE) [28]. The reconstruction on Nvidia GTX 4090 GPU by FBP algorithm of data with $N_x = N_z = 1000$ and $N_a = 500$ was performed in $T_{total} = 4s$. (Table 2).

Before we proceed with comparing the reconstruction and acquisition times, it should be noted that the classical form of CT reconstruction algorithms such as FBP may be far from optimal for producing partial reconstructions under MTR conditions. It is understandable

because the production of partial reconstruction is not a question of interest for classical CT by itself and has not been carefully studied yet. Although some obvious optimizations may easily be demonstrated in the example of FBP (and FDK) following from its additivity properties. In this type of algorithm, the value in each voxel of reconstruction may be calculated as

$$v(x, y, z)_n = \frac{\sum_{i=1}^{n} \widetilde{p}_i(x', z')}{n}, \tag{7}$$

where $\widetilde{p}(x', z')$ is a pixel of a filtered projection on a line connecting the X-ray source and voxel $(x, y, z)$ position and $n$ is a number of available projections. Next, it is possible to introduce a recurrent calculation method for $n + 1$ projection utilizing precalculated reconstruction for $n$ projections:

$$v(x, y, z)_{n+1} = \frac{n \times v(x, y, z)_n + \widetilde{p}_{n+1}(x', z')}{(n + 1)}. \tag{8}$$

The relation provided by Eq 8 allows calculation of a single partial reconstruction as $O(1)$ by total number of projections, while the standard method by the Eq 7 would require $O(n)$ operations. We will further refer to reconstruction methods based on Eq 8 with "partial" sub index, i.e FDK$_{partial}$.

The reconstruction times estimation for the optimized in proposed fashion algorithms is also listed in Table 3. This example demonstrates, that utilizing the previously obtained partial reconstructions may reduce required reconstruction times by orders of magnitude. It should be emphasized, that the proposed optimization concerns only the case of producing many reconstructions from the ordered subsamples of the same data. This task does not occur in classical CT and has not yet received the attention it deserves. The MTR protocol is the first case to our knowledge where such a task occurs, and we propose a simple and effective way of solving it. Applying a classical reconstruction approach to produce $N$ consecutive reconstructions from partial data would require back-projecting $N \times (N + 1)/2$ (Eq 7) projections, while the proposed optimization would require back-projecting of only $N$ projections (Eq 8). This optimization utilizing the basic additive properties of classical reconstruction algorithms, may change the way of thinking about the reconstruction process regarding MTR. Since the main concern MTR faces is often the computational burden of producing many reconstructions, here we demonstrate that in case of integral algorithms such as FBP and FDK these many reconstructions can be obtained for the time comparable to computing only a single reconstruction from a full data set, which is relatively fast. Similar kinds of optimizations should also be possible in the case of iterative algorithms since their convergence speed is strongly related to the accuracy of the initial approximation.

The main question under current consideration is if it is possible to achieve real-time reconstruction on current-generation setups, especially in the cases of very low acquisition

**Table 3. Maximum partial reconstruction time for different reconstruction algorithms and cases.**

| Parameter | micro CT | nano CT | medical CT | cargo CT |
|---|---|---|---|---|
| FBP, FDK, s. | 31.3 | 48.9 | 0.026 | 0.01 |
| SIRT, 50 iterations, s. | $4.5 \times 10^3$ | $7 \times 10^3$ | 3.3 | 1.45 |
| FBP$_{partial}$, FDK$_{partial}$, s. | 0.06 | 0.06 | $3.6 \times 10^{-5}$ | $1.5 \times 10^{-5}$ |
| Acquisition time, s. (Table 1) | $1.1 \times 10^3$ | $4.9 \times 10^4$ | 1 | 60 |

Estimations of maximum partial reconstruction time obtained from model Eq 6 and parameters from Table 1. For comparison total acquisition time is also reproduced from Table 1.

**Table 4. Number of partial reconstruction per full acquisition for different reconstruction algorithms.**

| Parameter | micro CT | nano CT | medical CT | cargo CT |
|---|---|---|---|---|
| FBP, FDK | 35 ($0.07 \times N_p$) | $10^3$ ($1.25 \times N_p$) | 38 ($0.05 \times N_p$) | $6 \times 10^3$ ($8.3 \times N_p$) |
| SIRT, 50 iterations | <1 | 7 ($0.008 \times N_p$) | <1 | 41 ($0.06 \times N_p$) |
| FBP$_{partial}$, FDK$_{partial}$ | $1.8 \times 10^4$ ($36 \times N_p$) | $8.2 \times 10^4$ ($10^3 \times N_p$) | $2.7 \times 10^4$ ($38 \times N_p$) | $4 \times 10^6$ ($5.5 \times 10^3 \times N_p$) |
| Acquisition time, s. (Table 1) | $1.1 \times 10^3$ | $4.9 \times 10^4$ | 1 | 60 |
| Max number of projections, $N_p$ (Table 1) | 512 | 800 | 720 | 720 |

Number of partial reconstruction per full acquisition (based on Table 3). The relation to the maximum number of projections is provided in parenthesis $N_p$ (Table 1). "<1" notation is used to note the cases in which real-time reconstruction becomes impossible. For comparison total acquisition time and maximum number of projections $N_p$ is also reproduced from Table 1.

times, typical for medical CT. To address this problem let us consider a relation between a reconstruction and acquisition times (Table 4). The Table 4 is produced by dividing the total experiment to partial reconstruction times from Table 3, with results also presented in relation to a total projection count from Table 1. The cases for which real-time reconstruction is not achievable are marked as "<1". The results in Table 4 demonstrate that real-time reconstruction is indeed possible not only for not-optimized classical algorithms but even for the use of iterative SIRT reconstruction in nano and cargo CT cases. It is also important to emphasize that optimized versions of algorithms exceed a theoretical resolution limit by orders of magnitude, creating time reserves, which could be put to use by constructing sophisticated quality assessment methods.

In the scope of the present paper the question of quality assessment methods to some extent is forcefully ignored, due to being very specific not only to the application area but also to a solving task and the method chosen for the solution. While general simple methods (i.e. calculating the norm of difference between consecutive partial reconstructions) may be much faster compared to the reconstruction process, sophisticated methods based on neural networks or forward projection calculations, may be comparable or even significantly slower than partial reconstruction times. The proposed model should be easily expanded on the case of slow quality assessment methods for a particular task with available estimation for typical processing times by including it in reconstruction times.

Based on proposed models and calculated estimations there are two promising combinations of reconstruction algorithms and acquisition orders for the cases of micro, nano, and cargo CT: (1) consecutive order with slow algorithms adequate to a limited angles case, (2) logarithmic order with optimized integral reconstruction algorithms. However, for the first case, the iterative partial reconstruction optimization should probably be researched closely. At the same time for the case of medical CT, the reconstruction times are even more critical and the first combination seems unrealistic even with possible additional optimizations. The second case is related to significantly slower acquisition (up to 5-6 times), which also lowers the potential for practical use.

## Experimental validation and practical realization of real-time reconstruction

In this section, the details of the practical validation of the proposed models are described. For this purpose, we constructed an experimental stand based on the available micro CT setup and control station. For experiment control and real-time reconstruction a SmartTomoEngine

**Table 5. The results of validation experiments for real-time reconstruction.**

| | Consecutive order, real-time | Logarithmic order, real-time | Consecutive order, standard |
|---|---|---|---|
| Number of projections, ($N_p$). | 512 | 512 | 512 |
| Measured total acquisition time, s. | $1.3 \times 10^3$ | $1.65 \times 10^3$ | $1.3 \times 10^3$ |
| Modeled total acquisition time, s. | $1.1 \times 10^3$ | $1.54 \times 10^3$ | $1.1 \times 10^3$ |
| Measured total reconstruction time, s. | $1.2 \times 10^3$ | $1.35 \times 10^3$ | - |
| Number of partial reconstructions ($N_{rec}$) | 385 | 440 | - |

The results of validation experiments for real-time reconstruction for consecutive and logarithmic acquisition orders. The time for standard protocol is also provided for comparison.

(STE) [28] software was used. The main parameters of the experimental setup and control station are provided in Table 5.

STE kernel allows coupling with the micro CT setup by custom protocols, achieving full control over the acquisition process. The experiment realization was purely software-based, which means no modifications were made to a micro CT setup control hardware. The control tools for setup parameters, acquisition, and reconstruction were connected via a graphical user interface (GUI). The GUI also allows the production of interactive real-time visualization of the latest produced partial reconstruction with both slice- and 3D-view. Based on visualization an operator is capable of manually interrupting the acquisition process. In this sense, the present setup could be considered a "manual MTR" protocol implementation. It should be noted that it is not yet a complete MTR, since it does not provide automatic quality assessment and does not implement stopping rules, but it is a very important step for MTR realization since it presents validation of the very possibility of practical implementation of MTR.

The flowchart in Fig 5 shows the principal scheme of the experiment. The acquisition and reconstruction processes are started simultaneously and asynchronously. The reconstruction utilizes the proposed optimized version of FDK$_{partial}$ based on Eq 8. Each captured projection is added to the reconstruction queue. The first projection triggers a reconstruction loop. After the current partial reconstruction is over all the projections from the queue are reconstructed.

We conducted two experiments with consecutive (Fig 2a) and logarithmic (Fig 2c) order. Times for each projection acquisition and each cycle of reconstruction are plotted in Fig 6a and 6b. The resulting total acquisition time and reconstruction counts for both cases, together with corresponding estimation from model Eq 4 are presented in Table 6. The total reconstruction time for all partial reconstruction is also provided to demonstrate that down-time for the reconstruction loop is reasonably small. The number of prescan projections for logarithmic order was chosen to be $N_{prescan} = 4$ projections.

It should be noted that at the point where the average projection acquisition time becomes lower than the average partial reconstruction cycle time (Fig 6b), the latter grows due to larger reconstruction queue sizes (Fig 5). In our realization, due to inner tools restrictions, reconstruction time is defined dominantly by loading in and out of GPU memory of reconstruction volume (which is much larger than queued projections memory size). Also, the volume renormalization (7) had to be performed on CPU, which should account for the discrepancies with the proposed model and experiments. This leads to partial reconstruction time being only slightly dependent on the count of projections used for reconstruction. This imperfection demonstrates another point of optimization: if possible the reconstruction volume should be kept in GPU memory between the partial reconstruction cycles, as it was assumed in the proposed model.

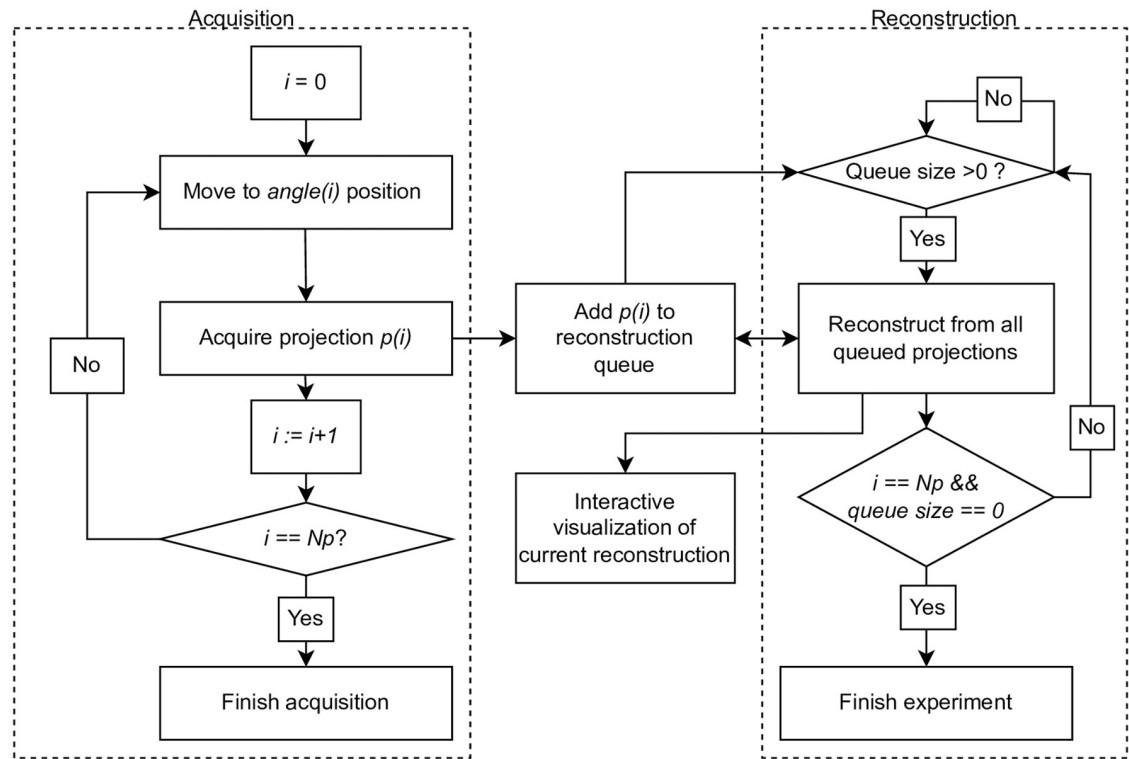

**Fig 5. Flowchart of real-time reconstruction implementation.**

The estimated experiment time corresponds well to the estimations provided by model Eq 4 (see Table 5). It should be emphasized that even with imperfect realization, for both cases real-time reconstruction was achieved on a typical consumer-grade GPU unit of *previous* generation. It should also be noted, that the proposed pipeline is already helpful by itself, besides

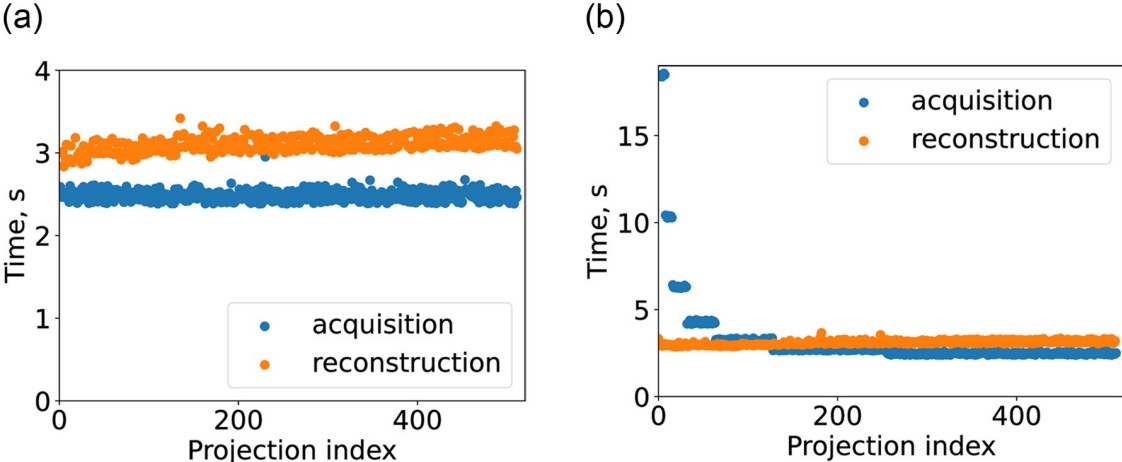

**Fig 6. The time required for the acquisition of a single projection and for performing a single partial reconstruction.** For reconstruction, the x-axis index is defined by the index of the first projection in the reconstruction queue. (a) Consecutive order. (b) Logarithmic order.

**Table 6. Parameters of the experimental stand.**

| Micro CT setup | Values | Control station | Values |
|---|---|---|---|
| Manufacturer | ELTEH-Med [5] | Software | SmartTomoEngine [28] |
| Sample rotation angles range | 360 | CPU model | AMD Ryzen 7 3700X 8-Core Processor 3.60 GHz |
| Total angles ($N_p$) | 512 | RAM size | 128Gb |
| Detector size ($N_x \times N_z$) | $2048 \times 2048$ | GPU model | Nvidia GTX 3080Ti |
| Frame exposition, s. ($T_{frame}$) | 1 | Floating point operations per second, TFlops/s. ($s_{rec}$) | 34.1 |
| Time per rotation, s. ($1/s_{rot}$) | 60 | Data loading speed throughout PCIe interface, MB/s. ($s_{load}$) | 500 |
| Detector overheads, s. ($C_{det}$) | 1 | Data access speed, TB/s. ($s_{memory}$) | 0.9 |
| Motion overheads, s. ($C_{move}$) | 0.1 | GPU RAM size, Gb | 12 |

The parameters of micro CT setup and control station used for validation experiments.

not being a complete MTR protocol. Even though it does not yet provide *automatic* quality control, it already allows for operator-based visual quality control, and the automatic quality control should be quite naturally incorporated within the proposed schema. Also, real-time reconstruction with consecutive angle order and proposed FDK$_{partial}$ optimization (Eq 8) is strictly faster than the total time needed to produce a resulting reconstruction utilizing standard protocol which includes synchronous acquisition and reconstruction. This happens due to acquisition time being the same in both cases, while in a later case, a full reconstruction should be produced afterward, as opposed to producing one final partial reconstruction in a real-time regime.

## Conceptual aspects of future generation CT utilizing MTR protocol

The software for the MTR itself is only to be created, but as it was demonstrated in the previous sections there should be no technical difficulties in the case of circular cone beam CT, since it requires only the wrapper software over the existing protocols and reconstruction algorithms. This means that for micro and nano CT application of MTR is possible on the current-generation setups with only software modifications.

At the same time, implementation of MTR in the areas of medical and cargo CT is met with certain problems, related to experiment time restrictions and design features of setups. The data in these modalities is acquired at a very high speed, produced by source-detector pair gantry rotation. It was shown (Table 1), that alternative acquisition orders would lead to a significant slowdown of an experiment. In this relation, the long-forgotten concept of electron beam computed tomography (EBCT) [27, 29] may surprisingly provide a solution to utilizing MTR in medical CT. EBCT allows to register projections from arbitrary angles while having only a small changeover overheads of $\approx 10 \mu s$. More so, EBCT may effectively utilize a pulse source of X-ray, which allows an artificial increase in the time between pulses without affecting the total dose. The last property is especially useful in the framework of MTR since it allows manipulation of the resolution on a decision scale. Similar to the previous section it is constructive to analyze the dependencies of dose and experiment time from the number of acquired projections (Fig 7). The model parameters for EBCT are also listed in Table 1.

It can be noted the dependencies for all protocols almost coincide, as to be expected since the effects of acquisition order are neglected. The absence of fast gantry rotation in EBCT is also profitable for MTR since it should be possible to locate the computing capacity extremely close to the detector, making use of fast data transition methods.

Thereby EBCT lifts the restriction of acquisition order for medical CT, allowing to utilization of fast reconstruction algorithms.

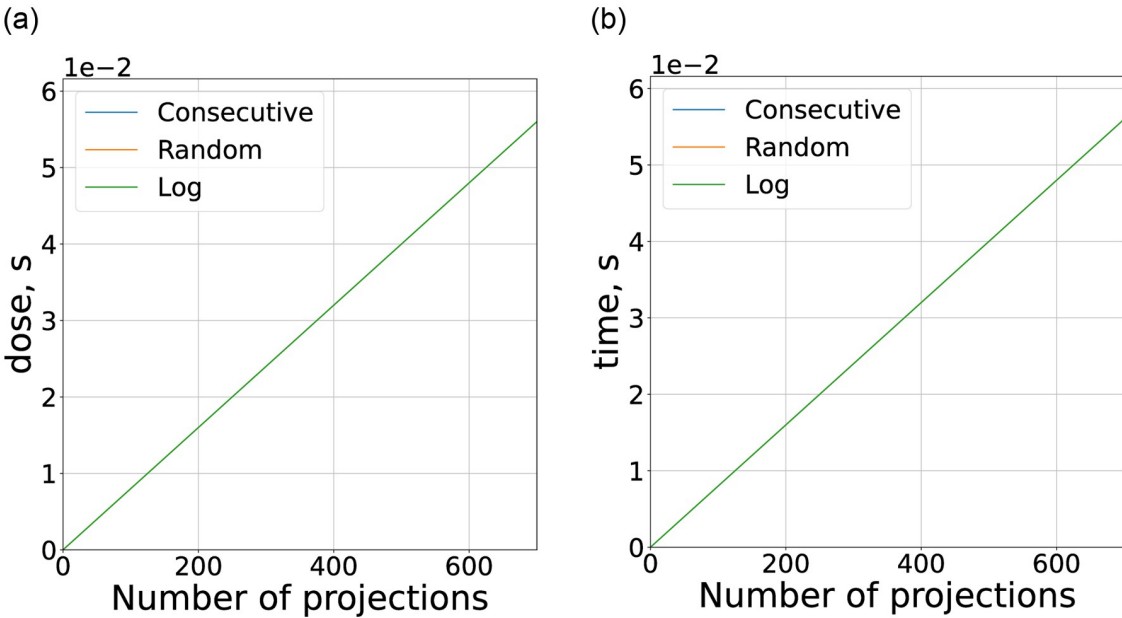

**Fig 7. Dose and experiment time dependence from the number of projections acquired with fixed maximum projection number** $N_p$ **based on model** Eq 3 **and with parameters from** Table 1**.** The dose is measured in object exposition time units for convenience of comparison. (a) Dose, EBCT. (b) Time, EBCT.

One of the main deductions that can be made based on estimations provided in the previous section is that the development of fast (in every sense) software and hardware systems, optimized for specific setups and studying goals, is crucial for the development of MTR. One of the main directions under this general goal is the development of fast iterative algorithms for partial reconstruction. From one point fast calculation of forward and backward projection operators is necessary, which recently got some progress from the side of methods based on fast Hough transform calculation [30–32], allowing to reach orders of magnitude speed up, relative to the classical approaches. From the other point, it is necessary to study the convergence rate of iterative algorithms utilizing previous partial reconstructions as a starting point.

The last point to discuss is the methods of quality assessment for early stopping decision making. It is expected that utilizing the methods that are specifically constructed for a certain task could substantially relax the requirements to the final reconstruction quality [14]. Quality assessment is directly connected with the concept of "early stopping", i.e. the acquisition interruption at the moment of detecting the fundamental problems with acquisition such as subject movement or implicit failure of some components. This concept would allow for decreased dose in the cases where in the classic CT multiple full dose experiments would be required. The concept of early stopping is promising for the universalization of MTR protocols, although the research in this area is yet to be started.

## Conclusion

The possibility and limits of MTR-based protocols application on current generation setups is studied in the presented paper. The numerical model for the estimation of dose and experiment time for different acquisition orders in MTR protocols is proposed. It is demonstrated, that random acquisition order leads to a major slowdown, being impractical for most of the real applications. An alternative acquisition protocol based on logarithmic order is proposed

and analyzed. It is shown that logarithmic order allows utilization of the classical algorithms for obtaining partial reconstructions while providing a compromise between rotation time overheads and partial reconstruction quality convergence. Also, a simple optimization of partial reconstruction production is introduced, which demonstrates orders of magnitude reconstruction time cost reduction.

Based on the proposed model the typical setups in medical, cargo, industrial, and nano CT areas of application are studied. It is shown that the most straightforward implementations of MTR are possible in the field of micro and nano CT. The realization of MTR on these types of setups does not require hardware enhancements and could be achieved by only providing specialized operating software. From the point of view of medical and cargo CT applications, the realization of MTR would probably require a new generation of equipment, with EBCT being a most promising modality for development lifting the restrictions on acquisition order. These results have to be taken in mind when constructing real MTR-based systems.

The proposed model is validated by implementing a real-time reconstruction protocol based on the existing commercially available micro CT setup and tomographic software SmartTomoEngine. It is demonstrated that real-time reconstruction is achievable even on previous-generation consumer-grade GPU units, with 385 and 440 partial reconstructions performed for consecutive and logarithmic order correspondingly for $N_p$ = 512 total projections available. The protocol based on real-time reconstruction utilizing consecutive order allows obtaining the final reconstruction insignificantly but strictly faster than the same acquisition based on standard protocol. For the logarithmic order with $N_p$ = 512 and $N_{prescan}$ = 4 acquisition is slowed by $\approx$ 22%, which is compensated by a much faster convergence of the partial reconstruction results. The measured experiment time is well approximated by the proposed model Eq 4.

It is also demonstrated that fast reconstruction algorithms suitable for an arbitrary acquisition order are crucial for MTR development. In this sense, the iterative reconstruction algorithms based on fast projection operators look the most promising [30–32]. The accessibility of fast reconstruction approaches could free the time resources for advanced quality assessment for early stopping, which would raise the generality of MTR approach.

## Author Contributions

**Conceptualization:** Marat Gilmanov, Konstantin Bulatov, Dmitrii Nikolaev, Vladimir Arlazarov.

**Data curation:** Oleg Bugai.

**Formal analysis:** Marat Gilmanov.

**Funding acquisition:** Marina Chukalina, Dmitrii Nikolaev, Vladimir Arlazarov.

**Investigation:** Marat Gilmanov.

**Methodology:** Marat Gilmanov.

**Project administration:** Anastasia Ingacheva, Dmitrii Nikolaev, Vladimir Arlazarov.

**Software:** Marat Gilmanov, Oleg Bugai.

**Supervision:** Konstantin Bulatov, Marina Chukalina, Dmitrii Nikolaev, Vladimir Arlazarov.

**Validation:** Marat Gilmanov, Oleg Bugai.

**Visualization:** Oleg Bugai.

**Writing – original draft:** Marat Gilmanov.

**Writing – review & editing:** Konstantin Bulatov, Anastasia Ingacheva, Marina Chukalina, Dmitrii Nikolaev, Vladimir Arlazarov.

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
