## [Decision Letter · Decision Letter 0]

4 Jun 2024

PONE-D-24-13721Applicability and potential of monitored reconstruction in computed tomographyPLOS ONE

Dear Dr. Gilmanov,

Thank you for submitting your manuscript to PLOS ONE. After careful consideration, we feel that it has merit but does not fully meet PLOS ONE’s publication criteria as it currently stands. Therefore, we invite you to submit a revised version of the manuscript that addresses the points raised during the review process.

We look forward to receiving your revised manuscript.

Kind regards,

Don S

Academic Editor

PLOS ONE

“This work was supported by the Russian Science Foundation, project no. 23-21-00524.”

“This work was supported by the Russian Science Foundation, project no. 23-21-00524.”

“This work was supported by the Russian Science Foundation, project no. 23-21-00524.”

Review comments 

Need a discussion on the proposed MTR concept for dose reduction in comparison with other international standards. In the article, it was mentioned to 'stop the projection acquisition based on prior knowledge.' Need to explain the bottleneck of prior knowledge and provide a detailed empirical formulation of reconstruction speed

**Comments to the Author**

1. Is the manuscript technically sound, and do the data support the conclusions?

Reviewer #1: Partly

2. Has the statistical analysis been performed appropriately and rigorously? 

Reviewer #1: No

3. Have the authors made all data underlying the findings in their manuscript fully available?

Reviewer #1: No

4. Is the manuscript presented in an intelligible fashion and written in standard English?

Reviewer #1: Yes

5. Review Comments to the Author

Reviewer #1: Applicability and potential of monitored reconstruction in computed tomography

The article represents the empirical quantitative model for calculating the acquisition and reconstruction times of MTR. It is demonstrated that projection acquisition order has a significant impact on the time and dose of tomographic experiments. To estimate the restrictions and scope of applicability for MTR four typical commercial setups are studied within a proposed model.

Comments

1. Abstract seems general, please mention the specific techniques and optimization methods the author adopted for the experiment. Also, quantify the proposed method values with existing methods in the abstract.

2. Figures are not embedded in the article. It is difficult to review the manuscript.

3. What is FDK? Author used two reconstruction algorithms such as SIRT, and FBP. This is compared with different CT cases. Kindly explain more about, what the improvements are done by authors in the proposed algorithm compared with typical methods.

4. The author has to include the algorithm and flow chart of the algorithm in the article.

5. Author mentioned the experiments and demonstration setups for the reconstructions. Those are missing in the article.

6. Please quantify the reconstruction parameters such as acquisition time with existing methods. It is helpful to understand the processing time of the scanning.

7. What are the other parameters to measure the quality of the reconstruction?

8. What are software and hardware used for the method?

6. PLOS authors have the option to publish the peer review history of their article (what does this mean?). If published, this will include your full peer review and any attached files.

---

## [Author Response · Author response to Decision Letter 0]

18 Jun 2024

The response to the reviewers comments for the article entitled “Applicability and potential of monitored reconstruction in computed tomography”

Reviewer 1.

The article represents the empirical quantitative model for calculating the acquisition and reconstruction times of MTR. It is demonstrated that projection acquisition order has a significant impact on the time and dose of tomographic experiments. To estimate the restrictions and scope of applicability for MTR four typical commercial setups are studied within a proposed model.

Comments

1. Abstract seems general, please mention the specific techniques and optimization methods the author adopted for the experiment. Also, quantify the proposed method values with existing methods in the abstract.

2. Figures are not embedded in the article. It is difficult to review the manuscript.

3. What is FDK? Author used two reconstruction algorithms such as SIRT, and FBP. This is compared with different CT cases. Kindly explain more about, what the improvements are done by authors in the proposed algorithm compared with typical methods.

4. The author has to include the algorithm and flow chart of the algorithm in the article.

5. Author mentioned the experiments and demonstration setups for the reconstructions. Those are missing in the article.

6. Please quantify the reconstruction parameters such as acquisition time with existing methods. It is helpful to understand the processing time of the scanning.

7. What are the other parameters to measure the quality of the reconstruction?

8. What are software and hardware used for the method?

RESPONSE

Thank you for providing detailed and constructive comments. Based on your concerns we decided to include our experiments which were previously considered to be out of the scope of the present paper, but now we see that it would be a great contribution to the discussion, which will support our conclusions. For this purpose we added a full new section named “Experimental validation and practical realization of real-time reconstruction” (see more in the list of changes and new manuscript version). We also made various changes to the manuscript to comply with raised concerns, making the paper both more clear and valuable.

We would like to also address the issue of data availability (3). We need to clarify that for reproducing our results and of the presented graphs by Eq.4,5,6 only the data contained in Tables 1,2 is necessary. We double-checked that all the data necessary is provided in corresponding tables. If some information still appears to be missing we would be glad to add it as well on further request.

Point-by-point response.

Q1. Abstract seems general, please mention the specific techniques and optimization methods the author adopted for the experiment. Also, quantify the proposed method values with existing methods in the abstract. 

Response: Thank you for a valuable comment. Following your advice we provided the necessary info in abstract, together with quantitative estimations.

Q2. Figures are not embedded in the article. It is difficult to review the manuscript.

Response: Unfortunately, the restrictions in the upload system did not let us embed images into the manuscript. For convenience we embedded the images into the tracked changes version of the manuscript.

Q3. What is FDK? Author used two reconstruction algorithms such as SIRT, and FBP. This is compared with different CT cases. Kindly explain more about, what the improvements are done by authors in the proposed algorithm compared with typical methods.

Response: FDK (Feldkamp, Davis and Kress algorithm [L. A. Feldkamp, L. C. Davis, and J. W. Kress, J. Opt. Soc. Am. A 1, 612-619 (1984)]) is another integral algorithm which is very similar to FBP, although in contrast to FBP it can be safely applied to the cone beam case. We are sorry for the inconvenience occurred in the internal edition process which led to a missing definition, we now added the necessary explanation in the main text. 

The proposed optimization concerns only the case of producing many reconstructions from the ordered subsamples of the same data. This task does not occur in classical CT and for this reason was not yet considered. Applying a classical reconstruction approach to produce N consecutive reconstructions from partial data would require backprojecting N*(N+1)/2 projections (eq. 7), while the proposed optimization would require backprojecting of only N projections (eq. 8). This optimization utilizing the basic properties of classical reconstruction algorithms may change the way of thinking about the reconstruction process regarding MTR. Since the main concern MTR faces is often the computational burden of producing many reconstructions, here we demonstrate that in case of integral algorithms such as FBP and FDK these many reconstructions can be obtained for the time comparable to computing only a single reconstruction from a full data set, which is relatively fast. 

We added the corresponding clarification to the main text as well.

Q4. The author has to include the algorithm and flowchart of the algorithm in the article.

Response: We agree that the flowchart of the MTR could be helpful for the understanding of this concept and navigating in the presented paper. We added a flowchart of a principal scheme of the MTR based experiment. We also added a flowchart in the new experimental section for real-time reconstruction pipeline.

Q5. Author mentioned the experiments and demonstration setups for the reconstructions. Those are missing in the article.

Response: The description of micro-CT setup and reconstruction station used for the validation of the empirical models are now separated into the individual section “Experimental validation and practical realization of real-time reconstruction”. We hope this section fully covers the requested information and contributes more to discussion.

Q6. Please quantify the reconstruction parameters such as acquisition time with existing methods. It is helpful to understand the processing time of the scanning.

Response: The existing methods basically can be described as two separate consecutive processes. First, the projection acquisition is performed by consecutive order (fig. 2a of the main article). This time is listed in table 1 (last row). Then a reconstruction is performed from full acquired data. This time is listed in table 3 (first two rows). The total experiment time will be the sum of reconstruction and acquisition times, which will be strictly greater than the time required for real-time reconstruction utilizing the same acquisition order. We are grateful for this comment since it produces another point of consideration for MTR, and now we added a corresponding discussion in the main text.

Q7. What are the other parameters to measure the quality of the reconstruction?

Response: The actual quality of the reconstruction will be dependent on the goal of CT study. Some of the examples are provided in the text, but with the rise of neural network models the image quality understanding could change significantly in certain areas of application. Some specific tasks require less than 10 projections to be completed [G. Herl et al., "Task-Specific Trajectory Optimisation for Twin-Robotic X-Ray Tomography," in IEEE Transactions on Computational Imaging, vol. 7, pp. 894-907, 2021, doi: 10.1109/TCI.2021.3102824.] while for others 1000 projections could be not enough. While these are extreme cases, it serves to show the range of understanding the image quality in different areas which is why we mostly avoid this concept in the present study. We added some discussion to make this point more clear in the main text.

Q8. What are software and hardware used for the method.

Response: We expanded the software and hardware description used for validation experiments in the new section “Experimental validation and practical realization of real-time reconstruction”. Based on the estimations and provided in the paper we argue that for the reconstruction part of MTR protocol latest generation consumer grade GPU unit (i.e. GTX 4090) should be enough for reconstruction not to become a bottleneck, considering proposed optimizations to the reconstruction process. 

The new validation experiments are conducted with typical consumer grade GPU (GTX 3080 Ti). The micro CT setup was the same one used to estimate the parameters of the model in section “Analysis of MTR implementation for commercially available setups''. From the software side for our estimations and experiments commercially available software SmartTomoEngine was used. While the software for the complete MTR protocol itself is only to be created, it is demonstrated that real-time reconstruction already could be implemented by only modifying the software part of the setup, at least for the setups with simple motion protocols characteristic for industrial and nano ct. We clarified these points in the main text as well.

List of changes

1. Abstract. Deleted text in lines 2-5 “However, up to now all the research in the MTR field is focused on dose reduction rather than implementation in real-life instruments. The main question discussed in the present paper is whether the implementation of MTR is already possible on current-generation setups or if it would require new-generation hardware instruments.”

2. Abstract. Added text in lines 5-7 “We are the first to study the issue of practical implementation of MTR protocols in current-generation real-life instruments.”

3. Abstract. Added text in lines 10-11 “The new alternative acquisition most suitable for MTR protocols is proposed.”

4. Abstract. Deleted text in lines 12-14 “Also, possible optimizations of the acquisition and reconstruction protocols suitable for MTR are introduced and studied.”

5. Abstract. Added text in lines 14-20 “We construct an experimental stand for achieving a real-time reconstruction, together with validation of the proposed acquisition time model. We demonstrate that real-time reconstruction may be implemented without slowing down an acquisition process. An optimization of reconstruction from partial data is proposed, which allowed the production of 385 and 440 reconstructions for standard and proposed acquisition orders correspondingly during a single acquisition of 512 projections.”

6. Abstract. Deleted text in line 23 “The future of MTR is discussed.”

7. Author summary. Added text in lines 5-6. “We present a real-time reconstruction based acquisition protocol implementation on current-generation micro CT setup and control station.”

8. Sec. Monitored tomographic reconstruction. Added text in lines 57-58. “The principal scheme of acquisition based on MTR protocol is summarized as a flowchart in Fig. 1.”

9. Sec. Monitored tomographic reconstruction. Added Fig.1, Flowchart of MTR protocol.

10. Sec. Monitored tomographic reconstruction. Added text in lines 78-84. “The quality assessment problem is vast, and with the rise of neural network models the image quality understanding could be changed even more if the reconstruction image is used only by the corresponding artificial intelligence system. For example, some specific tasks in Robo CT [17] require less than 20 projections to be completed, while in nano CT 1000 projections could be not enough [4]. While these are extreme cases, they show the range of understanding the image quality in different areas which is why we further discuss the image quality in a somewhat general sense.”

11. Sec. Monitored tomographic reconstruction. Added text in lines 93-94. “FBP (Filtered Back Projection [18]) or FDK (Feldkamp, Davis and Kress algorithm [19])“

12. Sec. Real time reconstruction. Added text in lines 312-314. “In the proposed model the loading of data to GPU memory is supposed to be performed only once, and GPU memory is considered to be large enough to store both volume and projection data.”

13. Sec. Real time reconstruction. Added text in lines 319. “(STE) [28]”

14. Sec. Real time reconstruction. Added text in lines 335 “by total number of projections”

15. Sec. Real time reconstruction. Added text in lines 336-337 “We will further refer to reconstruction methods based on Eq. 8 with “partial” sub index, i.e FDKpartial”

16. Sec. Real time reconstruction. Added text in lines 341-355 “It should be emphasized, that the proposed optimization concerns only the case of producing many reconstructions from the ordered subsamples of the same data. This task does not occur in classical CT and has not yet received the attention it deserves. The MTR protocol is the first case to our knowledge where such a task occurs, and we propose a simple and effective way of solving it. Applying a classical reconstruction approach to produce N consecutive reconstructions from partial data would require back-projecting N × (N + 1)/2 (Eq. 7) projections, while the proposed optimization would require back-projecting of only N projections (Eq. 8). This optimization utilizing the basic additive properties of classical reconstruction algorithms, may change the way of thinking about the reconstruction process regarding MTR. Since the main concern MTR faces is often the computational burden of producing many reconstructions, here we demonstrate that in case of integral algorithms such as FBP and FDK these many reconstructions can be obtained for the time comparable to computing only a single reconstruction from a full data set, which is relatively fast. ”

17. Added a full new section “Experimental validation and practical realization of real-time reconstruction” on lines 390-448, including 2 new tables (Table 5, 6) and 3 new figures (Fig 5, 6(a), 6(b)). The section describes new experiments based on realization of real-time reconstruction protocol on existing micro CT setup. See main text for details.

18. Sec. Conceptual aspects of future generation CT utilizing MTR protocol. Deleted text in lines 451-453 “As was demonstrated in the previous section, the conditions for the application of MTR in cases of micro and nano CT are already met in current-generation setups.”

19. Sec. Conceptual aspects of future generation CT utilizing MTR protocol. Added text in lines 453-458 “The software for the MTR itself is only to be created, but as it was demonstrated in the previous sections there should be no technical difficulties in the case of circular cone beam CT, since it requires only the wrapper software over the existing protocols and reconstruction algorithms. This means that for micro and nano CT application of MTR is possible on the current-generation setups with only software modifications.”

20. Sec. Conclusion. Added text in lines 519-530. “The proposed model is validated by implementing a real-time reconstruction protocol based on the existing commercially available micro CT setup and tomographic software SmartTomoEngine. It is demonstrated that real-time reconstruction is achievable even on previous-generation consumer-grade GPU units, with 385 and 440 partial reconstructions performed for consecutive and logarithmic order correspondingly for Np = 512 total projections available. The protocol based on real-time reconstruction utilizing consecutive order allows obtaining the final reconstruction insignificantly but strictly faster than the same acquisition based on standard protocol. For the logarithmic order with Np = 512 and Nprescan = 4 acquisition is slowed by ≈ 22%, which is compensated by a much faster convergence of the partial reconstruction results. The measured experiment time is well approximated by the proposed model Eq. 4.” 

21. Deleted section Acknowledgments with funding information.

---

## [Decision Letter · Decision Letter 1]

2 Jul 2024

Applicability and potential of monitored reconstruction in computed tomography

PONE-D-24-13721R1

Dear Dr. Gilmanov,

We’re pleased to inform you that your manuscript has been judged scientifically suitable for publication and will be formally accepted for publication once it meets all outstanding technical requirements.

Kind regards,

Don S

Academic Editor

PLOS ONE

---

## [Editor Report · Acceptance letter]

11 Jul 2024

PONE-D-24-13721R1 

PLOS ONE

Dear Dr. Gilmanov, 

I'm pleased to inform you that your manuscript has been deemed suitable for publication in PLOS ONE. Congratulations! Your manuscript is now being handed over to our production team.

Kind regards, 

on behalf of

Dr. Don S 

Academic Editor

PLOS ONE